# Redefining the Use of Regorafenib and Trifluridine/Tipiracil Without Bevacizumab in Refractory Metastatic Colorectal Cancer: Findings from the ReTrITA Study

**DOI:** 10.3390/cancers17122037

**Published:** 2025-06-18

**Authors:** Carlo Signorelli, Maria Alessandra Calegari, Annunziato Anghelone, Alessandro Passardi, Chiara Gallio, Alessandro Bittoni, Jessica Lucchetti, Lorenzo Angotti, Emanuela Di Giacomo, Ina Valeria Zurlo, Cristina Morelli, Emanuela Dell’Aquila, Adele Artemi, Donatello Gemma, Alessandra Emiliani, Marta Ribelli, Domenico Cristiano Corsi, Giulia Arrivi, Federica Mazzuca, Federica Zoratto, Mario Giovanni Chilelli, Marta Schirripa, Francesco Schietroma, Maria Grazia Morandi, Fiorenza Santamaria, Manuela Dettori, Antonella Cosimati, Rosa Saltarelli, Alessandro Minelli, Emanuela Lucci-Cordisco, Michele Basso

**Affiliations:** 1Medical Oncology Unit, S.Rosa Hospital, ASL Viterbo, 01100 Viterbo, Italy; 2Oncologia Medica, Comprehensive Cancer Center, Fondazione Policlinico Universitario Agostino Gemelli—IRCCS, 00168 Rome, Italy; 3Department of Medical Oncology, IRCCS Istituto Romagnolo per lo Studio dei Tumori (IRST) “Dino Amadori”, 47014 Meldola, Italy; 4Division of Medical Oncology, Policlinico Universitario Campus Bio-Medico, 00128 Rome, Italy; 5Medical Oncology, “Vito Fazzi” Hospital, 73100 Lecce, Italy; 6Medical Oncology Unit, Department of Systems Medicine, Tor Vergata University Hospital, 00133 Rome, Italy; 7IRCCS Regina Elena National Cancer Institute, 00144 Rome, Italy; 8Medical Oncology Unit, ASL Frosinone, 03100 Sora, FR, Italy; 9Medical Oncology, Isola Tiberina Hospital, Gemelli Isola, 00186 Rome, Italy; 10Oncology Unit, Department of Clinical and Molecular Medicine, Sant’ Andrea University Hospital, Sapienza University of Rome, 00189 Rome, Italy; 11UOC Oncologia, Ospedale Santa Maria Goretti, ASL Latina, 04100 Latina, Italy; 12Medical Oncology Unit, San Camillo de Lellis Hospital, ASL Rieti, 02100 Rieti, Italy; 13UOC Oncology A, Policlinico Umberto I, 00161 Rome, Italy; 14Experimental Medicine, Network Oncology and Precision Medicine, Department of Experimental Medicine, Sapienza University of Rome, 00185 Rome, Italy; 15Medical Oncology Department, Ospedale Oncologico Armando Businco, 09121 Cagliari, Italy; 16Medical Oncology Department, UO Oncologia Universitaria della Casa della Salute di Aprilia, 04011 Aprilia, LT, Italy; 17UOC Oncology, San Giovanni Evangelista Hospital, ASL RM5, 00019 Tivoli, RM, Italy; 18Medical Oncology Department, UO Oncologia, Ospedale San Paolo, ASL RM4, 00053 Civitavecchia, Italy; 19UOC Genetica Medica, Dipartimento di Scienze della Vita e Sanità Pubblica, Fondazione Policlinico Universitario Agostino Gemelli, IRCCS, 00168 Rome, Italy; 20Medical Oncology Department, Comprehensive Cancer Center, Fondazione Policlinico Universitario Agostino Gemelli, IRCCS, 00168 Rome, Italy

**Keywords:** regorafenib, trifluridine/tipiracil, metastatic colorectal cancer, sequencing, real-world evidence, survival, toxicity

## Abstract

The ReTrITA study is a comprehensive multicentre retrospective study examining real-world outcomes for regorafenib (R) and trifluridine/tipiracil (T) in patients with refractory metastatic colorectal cancer (mCRC). This study, conducted across 17 Italian centres from 2012 to 2023, analysed 1156 patients who underwent R and T as monotherapies or in sequential combinations (T/R and R/T) without bevacizumab. The primary goals were overall survival (OS) and progression-free survival (PFS), with secondary endpoints being the objective response rate (ORR), disease control rate (DCR), and toxicity. There was no significant difference in OS or PFS between the R and T monotherapies. Patients undergoing the R/T sequence experienced significantly better results compared to T/R, with an increased median OS (16.6 vs. 12.6 months; *p* = 0.0004) and PFS (11.5 vs. 8.5 months; *p* < 0.0001). These results were similar across clinically important subgroups, especially among elderly patients, those with RAS mutations, and those with microsatellite-stable tumours. Toxicity profiles differed between regimens: haematologic toxicity was more common with T, whereas non-haematologic adverse events predominated with R. The R/T sequence led to more grade 3/4 toxicities but less neutropenia. ReTrITA presents compelling real-world evidence for regorafenib as the preferred initial drug in sequential therapy. These findings suggest a tailored approach to salvage treatment for mCRC while also highlighting the need for future prospective trials that include emerging agents and combination strategies.

## 1. Introduction

Colorectal cancer (CRC) is the fourth most prevalent malignancy and the second major cause of cancer-related death in the United States. In 2018, it was the second leading cause of cancer-related deaths in Europe; however, mortality rates have been decreasing since 2012. In 2025, it is expected to account for roughly 154,270 new cases (7.6% of all newly diagnosed malignancies) and 52,900 deaths (8.6% of all cancer-related deaths), with a predicted 5-year relative survival rate of 65.4% [1,2]. Metastatic progression occurs in approximately 50–60% of patients with CRC, with the liver being the most common location of metastasis [3,4]. Among them, 80–90% have unresectable liver lesions [5]. Although liver metastases typically develop metachronously after primary tumour treatment [6], synchronous liver metastases occur in 20–34% of patients at the time of initial diagnosis [7].

The therapeutic landscape for metastatic colorectal cancer (mCRC) is highly complicated, needing long-term collaborative efforts to improve therapy options in a setting with many different clinical and molecular challenges [8].

Tumours with high microsatellite instability (MSI-H) have seen the most substantial therapeutic advances among the molecular subtypes of CRC, notably with the introduction of immune checkpoint inhibitors. Pembrolizumab’s efficacy in this subgroup has been established to have a significant impact on patient clinical care, both as a first-line treatment and in later therapeutic settings [9,10]. Nonetheless, MSI-H tumours account for only about 15% of all CRC cases, limiting the effectiveness of immunotherapy to a restricted subset of patients. As a result, significant research efforts are currently being directed towards expanding immunotherapy’s therapeutic potential to microsatellite-stable (MSS) tumours—a goal that, despite encouraging preclinical and early clinical findings, has yet to be fully realised.

An important therapeutic issue is managing severely pretreated mCRC patients who still have satisfactory functional status but have limited access to additional targeted therapies—either because of a lack of available alternatives or exclusion from clinical trials. In such circumstances, therapeutic decision-making for third-line treatment and beyond is influenced by a number of fundamental criteria, including prior treatments, tumour biology, patient performance status, and, most importantly, the anticipated toxicity of proposed drugs, which has a considerable impact on quality of life. There is currently no universally accepted standard or recommended salvage therapy sequence. While patients with targetable mutations should be referred for suitable therapies, it is becoming increasingly unusual for RAS wild-type patients to have not previously had anti-epidermal growth factor receptor (EGFR) therapy or for MSI-H patients to have not received immunotherapy. Rechallenge approaches are frequently attempted, but their effectiveness can frequently be compromised by cumulative toxicity and reduced bone marrow reserve [11,12,13,14,15]. Mutation-driven treatments for HER2, NTRK, or RET are rarely used and are largely limited to clinical trials [16,17,18,19,20,21].

Currently, the main drugs licensed for use in refractory mCRC are trifluridine/tipiracil (T) (a combination of trifluridine, a thymidine-based nucleoside analogue, and tipiracil, which is a thymidine phosphorylase inhibitor), fruquintinib (a selective inhibitor of vascular endothelial growth factor receptors (VEGFR) 1, 2, and 3) (pending reimbursement in some countries, such as Italy), and regorafenib (R) (a multi-kinase inhibitor targeting VEGFR, FGFR, PDGFR, KIT, RET, etc.). Trifluridine/tipiracil has also been used in combination with bevacizumab, with good outcomes [22,23,24,25,26]. However, the optimal sequencing of T and R is unknown in the absence of direct-comparison investigations.

To address this unmet need, the ReTrITA (Regorafenib and Trifluridine/Tipiracil in Italian mCRC Patients) study was designed as a retrospective, multicentre real-world analysis to assess outcomes associated with the use of R and T, administered sequentially and non-sequentially—and, importantly, without bevacizumab—in heavily pretreated mCRC patients. Building on our previous large retrospective experience [27], this study proposes providing real-world evidence on treatment patterns, safety profiles, treatment duration, and survival outcomes, thereby adding to a better understanding of sequencing approaches in clinical practice.

The findings are intended to contribute to the current research on salvage therapy sequencing in mCRC, filling a major vacuum left by pivotal trials such as CORRECT and RECOURSE [22,23], which assessed R and T separately but did not analyse their comparative or sequential use. Furthermore, the result adds to recent data, notably those from the SUNLIGHT trial [26], which showed better efficacy when T was administered in combination with bevacizumab, showing an ongoing development of post-progression therapy regimens in mCRC.

## 2. Patients and Methods

This 11-year retrospective observational study was carried out at 17 cancer centres in Italy between 2012 and 2023. The following criteria were used for inclusion: histologically confirmed stage IV adenocarcinoma of the colon or rectum with unresectable metastatic disease; age ≥ 18 years; Eastern Cooperative Oncology Group Performance Status (ECOG PS) of 0 to 2; documented RAS mutation status; and adequate organ function at the start of treatment; documented progression of the disease after at least two previous lines of standard chemotherapy, including fluoropyrimidines, irinotecan, oxaliplatin, anti-vascular endothelial growth factor (VEGF) antibodies (bevacizumab or aflibercept), and/or anti-EGFR antibodies (cetuximab or panitumumab).

If patients were given maintenance therapy after establishing disease control, the induction of therapy and the administration of a maintenance regimen would be considered a single therapeutic approach. Patients who had previously had a first-line irinotecan regimen were able to receive a single-agent irinotecan as a third-line treatment.

Patient gender, age at third-line therapy initiation, surgical history, molecular characteristics (RAS/BRAF mutation status, mismatch repair status, and/or microsatellite instability), primary tumour location, metastatic sites, and prior chemotherapy and targeted agent exposure were among the baseline variables. Additional clinical data were collected retrospectively, including ECOG PS, start and discontinuation dates for R or T, reasons for treatment discontinuation, best tumour response, progression-free survival (PFS), overall survival (OS), dose modifications, and treatment-related grade 3/4 adverse events (AEs) classified according to the National Cancer Institute Common Terminology Criteria for Adverse Events (CTCAE), version 4.0.

This study was approved by the Ethics Committee of Area 4 Lazio in Rome, Italy (protocol number 29-2024, approved on 4 March 2024) and carried out in conformity with the Declaration of Helsinki. To protect patient anonymity, all data were anonymised, with patients identified just by initials and a numerical code. In accordance with data protection requirements, the main investigator served as the designated data manager, having exclusive access to the entire dataset. In conformity with ethical norms and the retrospective character of the investigation, informed consent was waived in cases where patients could not be contacted or refused to provide consent. Patients were monitored until they died or lost contact. The study design is illustrated in Figure 1.

### 2.1. Outcome Parameters

This study’s primary endpoints were overall survival (OS) and progression-free survival. OS was defined as the time from the start of treatment until death from any cause. For patients in the sequential therapy cohorts (R/T or T/R), OS was computed from the start of the first agent (R or T) until death occurred during or after the second agent (T or R). PFS was defined as the time from the beginning of treatment until either confirmed progression of the disease or death from any cause, whichever occurred first. PFS in the sequential treatment arms was calculated from the start of the first agent (T or R) until the time of progression or death during or after the second agent (R or T). Patients were censored at the last date they were known to be alive if they were alive and not experiencing any progression at the time of the data cut-off.

The objective response rate (ORR), which is the percentage of patients who achieve either CR or PR, and the disease control rate (DCR), which is the percentage of patients who achieve a complete response (CR), partial response (PR), or stable disease (SD), as determined by RECIST version 1.1, were secondary endpoints. DCR and ORR were examined by the treatment setting, comparing the outcomes for patients receiving monotherapy and sequential therapy.

To minimise selection bias, the final analysis comprised all patients who received R, T, or sequential R/T or T/R. According to the law, the lead investigator, who was also the data manager, had complete access to the whole database and carried out the statistical analysis, while designated investigators who were blinded to clinical outcomes took care of patient selection.

To further reduce the risk of bias, all outcome measures and definitions were predetermined before data collection. However, given this study’s retrospective nature, the findings should be considered to be exploratory.

### 2.2. Drug Administration

Patients received either R or T based on conventional clinical practice and the treating physician’s discretion. To minimise toxicity and adhere to real-world clinical standards, clinicians were permitted to adopt dose reductions or delays based on patient tolerance and the occurrence of adverse events. Regorafenib was given orally at a starting dose of 160 mg once daily for the first 21 days of each 28-day cycle. The ReDOS dose-escalation method for R was used in selected patients at the discretion of the physician [28]. This strategy entailed starting therapy at 80 mg/day, with weekly dose increments of 40 mg, up to a maximum of 160 mg/day, if no substantial treatment-related adverse effects occurred. Trifluridine/tipiracil was given orally at a dose of 35 mg/m^2^ twice daily on days 1–5 and 8–12 of each 28-day cycle, with a 14-day rest interval. Dose changes for T were also approved in the event of clinically substantial toxicity. Treatment with either of the drugs was maintained until disease progression, intolerable toxicity, deterioration of performance status, patient refusal, or physician decision. In sequential therapy groups (T/R or R/T), clinicians were responsible for selecting and timing the second-line drug based on patient status, historical response, and safety considerations.

### 2.3. Statistical Analysis

All statistical analyses were performed with MedCalc for Windows, version 19.4 (MedCalc Software, Ostend, Belgium). Descriptive statistics were applied to summarise baseline demographic, clinical, and treatment parameters. Fisher’s exact test or a chi-square test was used to investigate associations between categorical variables, as applicable. The Kaplan–Meier method was used to estimate PFS and OS, with the log-rank test performed to examine subgroup differences. A two-sided *p*-value ≤ 0.05 was determined to indicate statistical significance.

To investigate differences in treatment efficacy across clinically relevant subgroups, an exploratory subgroup analysis was performed. The subgroups included age, sex, ECOG PS, RAS mutation status, mismatch repair (MMR) status, primary tumour location (right colon, left colon, or rectum), prior exposure to anti-EGFR and anti-VEGF agents in the frontline setting, history of adjuvant treatment, use of rechallenge strategies, and metastatic disease pattern. Forest plots were utilised to graphically present hazard ratios and 95% confidence intervals for OS and PFS across various subgroups in order to help with comparison interpretation.

## 3. Results

### 3.1. Patient Characteristics

Table 1 summarises the baseline demographic and clinical characteristics of patients enrolled in the ReTrITA study, divided by treatment group (T/R vs. R/T) and monotherapy group (T vs. R). A total of 1156 patients were assessed: 261 received sequential trifluridine/tipiracil followed by regorafenib (T/R), 155 received the inverse sequence (regorafenib followed by trifluridine/tipiracil, R/T), 427 received trifluridine/tipiracil monotherapy (T), and 313 received regorafenib monotherapy (R).

The median age across treatment groups was comparable, with no statistically significant variations. The T group exhibited a greater proportion of patients aged ≥70 years than the R group (55.3% vs. 36.1%, *p* < 0.0001). The distribution of sex was uniform among groups.

The RAS mutation status varied considerably between monotherapy groups (*p* = 0.0119), with a greater percentage of wild-type tumours in the R group. The primary tumour location varied significantly (*p* < 0.0001), especially in rectal cancer prevalence, which was lower in the R group. Significant differences (*p* < 0.0001) were found between the T and R groups for MSI status and MMR proficiency. The R group had a greater number of patients with an unknown or untested MMR status.

ECOG performance status, prior adjuvant chemotherapy exposure, and metastatic disease distribution were broadly balanced among the sequential treatment groups, but there were some differences between monotherapy groups. Notably, previous adjuvant therapy was more common in the T group than in the R group (*p* = 0.0022).

The monotherapy cohorts differed significantly in the regimens of chemotherapy they received for first- and second-line treatment. Earlier in the treatment pathway, patients in the T group were more likely to have been administered biological therapies and more rigorous regimens. Compared to patients in the R group, a significantly greater percentage of patients in the T group received anti-EGFR and anti-VEGF antibodies in both first and second lines (*p* < 0.0001 for both lines).

The R group was more likely than the T group to be given rechallenge therapy using previously administered drugs (*p* = 0.0151). These results point to some variations in baseline features and previous treatment exposures that could affect the sequencing and outcomes of therapy.

### 3.2. Survival Outcomes in the Sequential Treatment Groups

Patients receiving sequential therapy T/R (*n* = 261) compared to the reverse sequence (R/T, *n* = 155) were evaluated for OS and PFS, assessed by applying Kaplan–Meier analyses. With a median time period of 16.6 months (95% CI: 14.8–101.0), as opposed to 12.6 months (95% CI: 11.1–14.3) in the T/R group, the R/T sequence showed a statistically significant improvement in OS (HR = 0.67; 95% CI: 0.53–0.83; *p* = 0.0004). The R/T and T/R groups had estimated 2-year OS rates of 27.7% and 16.8%, respectively (Figure 2).

Similarly, the median PFS for the R/T cohort was 11.5 months (95% CI: 13.4–17.6), while that for the T/R group was 8.5 months (95% CI: 9.9–11.6) (HR = 0.60; 95% CI: 0.49–0.74; *p* < 0.0001). In this real-world scenario, the clinical benefit of starting regorafenib before trifluridine/tipiracil was highlighted by the one-year PFS rates of 45.5% for R/T and 27.6% for T/R (Figure 3).

### 3.3. Survival Outcomes in the Monotherapy Groups

Overall survival (OS) and PFS results were similar across treatment arms in the patient groups who did not receive sequential treatment with T and R.

With 361 and 286 events recorded, respectively, the median OS for the T and R groups was 5.9 months (95% CI: 3.5–7.7) and 5.0 months (95% CI: 9.0–13.4). The R group produced a somewhat greater 2-year OS rate (7.9%) than the T group (3.5%). However, there was no statistically significant difference in the survival outcomes between the two monotherapies, as indicated by the calculated hazard ratio (HR) for OS, which was 1.01 (95% CI: 0.86–1.19; *p* = 0.8371) (Figure 4).

Similarly, with 399 and 308 events recorded, respectively, the median PFS for T and R was 3.3 months (95% CI: 4.0–4.8) and 3.2 months (95% CI: 3.8–4.9), respectively. The T group’s 1-year PFS rate was 6.0%, while that of the R group was 4.5%. There was once more no discernible difference between the two therapies, as evidenced by the HR for a PFS rate of 1.03 (95% CI: 0.88–1.20; *p* = 0.6531) (Figure 5).

These findings indicate that, in the absence of a sequential treatment approach, T and R produce comparable clinical outcomes in terms of OS and PFS. The lack of statistically significant differences supports the notion that both treatments might be regarded as an effective choice in this late-line scenario when administered alone.

### 3.4. Summary of Efficacy Outcomes

The results of this study’s efficacy, as outlined in the preceding paragraphs, are succinctly summarised in Table 2. In addition, the objective response rate (ORR) was modest across all groups but numerically greater in the T/R group (3.4%) than in the R/T group (1.0%), albeit the difference was not statistically significant (*p* = 0.9564). However, the disease control rate (DCR), which pertains to stable disease in addition to ORR, was considerably higher in the R/T group (52.1%) compared to the T/R group (32.9%) (*p* = 0.9564) and markedly better than in the single-agent arms (T: 23.5%, R: 22.9%, *p* < 0.0001).

These findings suggest the increased efficacy of sequential treatment, particularly the R/T sequence, for both survival and disease control.

Statistical analysis revealed a significant survival advantage in the R/T group over other treatment options. The hazard ratio (HR) for OS in the T/R group compared to R/T was 0.67 (95% CI: 0.53–0.83, *p* = 0.0004), and for PFS, it was 0.60 (95% CI: 0.49–0.74, *p* < 0.0001), indicating a considerable advantage for the R/T sequence.

### 3.5. Comparative Analysis of Subgroup Survival in the Sequential Treatment Groups

A comprehensive subgroup analysis was performed to investigate characteristics related to clinical benefits among patients receiving sequential treatment T/R vs. the reverse sequence. Forest plots were created to evaluate OS and PFS across a variety of clinically relevant subgroups (Figure 6).

The R/T sequence significantly improved OS and PFS in patients aged ≥70 years compared to T/R (OS HR = 0.58; 95% CI: 0.41–0.81; *p* = 0.0017; PFS HR = 0.48; 95% CI: 0.34–0.66; *p* < 0.0001). Patients <70 years old showed a similar trend; however, it was not statistically significant.

Among ECOG PS categories, patients with PS 1 exhibited the most evident benefit with R/T (OS HR = 0.60; 95% CI: 0.45–0.80; *p* = 0.0007; PFS HR = 0.58; 95% CI: 0.44–0.76; *p* < 0.0001). PS 0 patients also demonstrated favourable results in PFS (HR = 0.65; 95% CI: 0.46–0.91; *p* = 0.0147). There were no significant differences detected for PS 2.

Patients with mutant RAS showed a higher benefit from the R/T sequence for OS and PFS (HR = 0.59 and 0.57, respectively; *p* < 0.001). There was a trend towards increased survival in wild-type RAS patients, but the results were not statistically significant.

In the MSS subgroup, R/T was substantially related to improved OS (HR = 0.57; 95% CI: 0.43–0.76; *p* = 0.0001), while there was no apparent benefit in MSI patients.

Patients who previously received anti-VEGF therapy experienced significantly better OS and PFS with R/T (OS HR = 0.73; *p* = 0.0448; PFS HR = 0.65; *p* = 0.0029). Previous treatment with anti-EGFR antibodies was likewise associated with longer PFS (HR = 0.62; *p* = 0.0168).

Patients with liver metastases (alone or in combination with other disease sites) experienced a significant improvement in both OS and PFS when treated with the R/T sequence. There was no significant difference in OS across patients who had further metastatic locations; however, PFS remained higher for R/T patients.

Patients with right-sided tumours had a better OS and PFS with R/T (OS HR = 0.67; *p* = 0.0461; PFS HR = 0.53; *p* = 0.0006), while a similar trend was observed for left-sided and rectal cancers.

The R/T sequence improved outcomes for both male and female patients, with females benefiting more than men (OS HR = 0.57; *p* = 0.0025; PFS HR = 0.54; *p* = 0.0003).

Patients who had not received prior rechallenge therapy or adjuvant chemotherapy exhibited more substantial benefit from R/T in both OS and PFS (all *p* < 0.001), while the effect was not as evident in those who had received these therapies.

Age, sex, ECOG performance status, RAS mutational status, mismatch repair status, tumour sidedness (right vs. left colon vs. rectal cancer), prior exposure to anti-EGFR or anti-VEGF agents, and prior adjuvant therapy are all presented with hazard ratios (HRs) and 95% confidence intervals (CIs). The R/T sequence is favoured when the HR is less than 1. The investigations were exploratory in nature, using real-world data from the past. While not all subgroup differences were statistically significant, the forest plots show a tendency favouring the R/T sequence in several clinically important groupings (Figure 6).

### 3.6. Comparative Analysis of Subgroup Survival in the Monotherapy Groups

Forest plot analyses were used for exploring potential subgroup differences in OS and PFS among patients receiving non-sequential T or R. Cox proportional hazards models were used to generate hazard ratios (HR) with a 95% confidence interval (CI), and the log-rank test was used to determine the statistical significance of intergroup differences (Figure 7).

Except for certain subgroups indicated below, no statistically significant difference in OS or PFS was found between T and R across all examined subgroups, including age, ECOG performance status, RAS status, metastatic sites, sex, prior biologic therapy, MMR status, primary tumour location, use of rechallenge therapy, and adjuvant therapy.

The overall log-rank test for OS across ECOG strata was significant (*p* = 0.0026), indicating performance status as a key prognostic factor. Notably, a longer median OS was observed among patients with ECOG PS 0 (7.7 months for T vs. 6.5 months for R; HR 1.00, 95% CI: 0.70–1.44; *p* = 0.9595). A statistically significant difference for PFS was found in the ECOG PS 2 subgroup (HR 1.67, 95% CI: 1.12–1.94; *p* = 0.0108), indicating that R worked unfavourably in this group.

Rechallenge therapy was found to have a survival advantage in terms of OS; patients who received rechallenge had significantly better OS when treated with T as opposed to R (HR 0.61, 95% CI: 0.39–0.96; *p* = 0.0332). This outcome might demonstrate the possible advantages of reintroducing the medication to properly chosen patients. However, this advantage was not shown for PFS.

There were trends towards better outcomes in some categories (e.g., patients undergoing adjuvant therapy appeared to derive somewhat better OS from T; HR 1.02, *p* = 0.9002), but these were not statistically significant, even if no significant differences were observed for other subgroups.

These findings support tailored treatment approaches in refractory mCRC by indicating that although overall treatment efficacy between T and R appears comparable in non-sequential administration, some clinical subgroups—such as ECOG PS 2 patients and those undergoing rechallenge therapy—may benefit more from particular treatment choices. However, these results should be regarded as hypothesis-generating and should be validated prospectively because of the analysis’s retrospective nature.

## 4. Safety

Grade 3/4 (G3/G4) toxicities were assessed in 426 patients who were sequentially treated with T without bevacizumab and R. Of them, 184 individuals were given the R/T sequence, while 242 patients were given the T/R sequence. In parallel, 234 patients receiving T and 155 receiving R had their toxicities from single-agent treatments evaluated. The toxicities associated with the drugs used in this study are compiled in Table 3.

In all groups, the incidence of any G3/G4 adverse event was 100%. However, compared to the T/R group, a significantly greater percentage of patients in the R/T group (69.0% vs. 55.5%; *p* = 0.0167) experienced at least one G3/G4 toxicity. On the other hand, patients treated with T experienced toxicity more often than those treated with R (41.4% vs. 37.3%; *p* = 0.0005).

The T/R group experienced more haematologic G3/G4 events than the R/T group (51.2% vs. 46.2%; *p* = 0.0070). On the other hand, haematologic toxicity was substantially higher with single-agent T than with R (77.8% vs. 8.4%; *p* < 0.0001). The most common haematologic event, neutropenia, was seen in 72.6% (T/R), 69.4% (R/T), 69.8% (T), and 30.8% (R) of cases. Compared to T (26.4%), anaemia was more common with R monotherapy (61.5%).

In the T/R and R/T groups, non-haematologic G3/G4 toxicities were reported in 48.8% and 53.8% of patients, respectively (*p* = 0.1971). However, the incidence of R monotherapy was much higher than that of T (91.6% vs. 22.2%; *p* < 0.0001). R-treated patients had higher rates of skin problems, hypertension, and hand–foot skin reaction.

Sequential treatment with R followed by T (R/T) was associated with a higher overall incidence of G3/G4 toxicities compared to T/R, primarily driven by non-haematologic events. In contrast, haematologic toxicity predominated in single-agent T treatment. These findings underscore the distinct toxicity profiles of T and R and support the need for tailored toxicity monitoring based on treatment sequence and regimen.

Non-haematologic events were the main cause of the greater overall incidence of G3/G4 toxicities linked to sequential treatment with R followed by T (R/T) as opposed to T/R. On the other hand, single-agent T therapy was dominated by haematologic toxicity. These results highlight the different toxicity profiles of T and R and provide support for the necessity of personalised toxicity monitoring according to treatment regimen and sequence.

## 5. Discussion

The ReTrITA study provides useful real-world data on the use of R and T, alone or in combination, in a large cohort of Italian patients with metastatic colorectal cancer (mCRC). This study examines the efficacy and safety profiles of these therapies, shedding light on their prospective roles in the mCRC therapeutic landscape. Furthermore, the data may assist doctors in making carefully considered choices about therapy sequencing in this difficult patient population. The information from the ReTrITA investigation, in particular, may provide a more individualised therapeutic strategy, guaranteeing that patients receive optimal care throughout the course of their disease. Continuous research and cooperation will therefore be essential to enhancing the prognosis of mCRC patients.

In the monotherapy context, ReTrITA showed no statistically significant difference in OS or PFS between patients who received R or T alone, which is consistent with the CORRECT and RECOURSE trials [22,23], as well as other real-world studies [29,30,31]. These data support the idea that both drugs are acceptable choices for salvage therapy in patients who are no longer eligible for conventional therapies. The sequential analysis, however, showed that the R/T sequence was significantly superior to T/R in terms of survival, both OS (16.6 vs. 12.6 months; HR = 0.67; *p* = 0.0004) and PFS (11.5 vs. 8.5 months; HR = 0.60; *p* < 0.0001). The results were robust because this survival advantage held true for clinically significant subgroups, such as patients with ECOG PS 1, RAS-mutated tumours, and microsatellite-stable disease. These results are significant since they are consistent with those of our previous real-world analysis (2023), which, in a large retrospective multicentre Italian investigation, similarly showed that the R/T sequence was statistically superior. In comparison to the T/R group, the R/T sequence generated a longer median OS (15.9 vs. 13.9 months, *p* = 0.0194) and PFS (11.2 vs. 8.8 months, *p* = 0.0005) in their cohort [27]. Significantly, the fact that these results were consistent between the two studies supports the hypothesis that R should be used initially in salvage situations.

The optimum sequencing of R and T in mCRC has received increased attention, particularly in patients with extensively pretreated disease. In this context, we compare our findings to four essential studies: PRODIGE 68 (SOREGATT), SEQRT2, the Ahn et al. study, and OSERO, all of which provide crucial and complementary insights into this subject [29,30,31,32].

The PRODIGE 68 (SOREGATT) randomised phase II trial is the only prospective study that directly assesses treatment feasibility using sequencing. The trial randomly assigned 234 individuals to either R/T (arm A) or T/R (arm B). The primary endpoint—the proportion of patients able to complete at least two cycles of both drugs—was considerably higher in the T/R arm (56% vs. 40%; *p* = 0.018), indicating that starting treatment with T may prolong clinical status and improve treatment completion. Furthermore, 39% of patients in the R/T arm did not progress to second-line therapy, owing to clinical deterioration [32].

Compared to PRODIGE, four observational studies—SEQRT2, OSERO, the Ahn et al. study, and ReTrITA—provide differing insights into efficacy based on sequence: SEQRT2, a real-world research of 308 patients in the United States, found that the R/T group had a higher median overall survival (12.8 vs. 10.2 months) and time to next treatment (TTNT: 9.3 vs. 8.6 months) than the T/R group. However, the difference was not statistically significant (HR for OS = 1.20; *p* = 0.20) [29]. In a study of 818 patients in U.S. community practice, Ahn et al. found that the R/T sequence resulted in a numerically improved OS in both third- and fourth-line settings (13.1 vs. 11.5 months and 11.6 vs. 10.3 months, respectively), but this finding was not statistically significant [30]. Our ReTrITA study, a large Italian retrospective cohort of 1156 patients, presented the strongest evidence in favour of the R/T sequence. This study found that R/T had a substantial advantage in terms of OS (16.6 vs. 12.6 months; HR = 0.67; *p* = 0.0004) and PFS (11.5 vs. 8.5 months; HR = 0.60; *p* < 0.0001). In contrast, the OSERO study, a Japanese prospective observational study, found no statistically significant difference in survival rates among patients starting with R, T, or T + bevacizumab. The median OS was 11.8 months (R-first), 7.1 months (T-first), and 10.2 months (T + bevacizumab-first), with adjusted hazard ratios showing no evident superiority after covariate adjustment [31]. ReTrITA, the Ahn et al. study, and SEQRT2 all showed that patients who received R before T had lower rates of neutropenia and required fewer myelosuppression-related treatments, including erythropoietin and G-CSF. For example, 26%/12% of R/T patients and 32%/16% of T/R patients, respectively, exhibited moderate/severe neutropenia in the Ahn et al. study. This pattern demonstrates that starting with R is haematologically tolerable, which is crucial for patients whose bone marrow reserves are impaired. Despite having a randomised design and strong internal validity, PRODIGE 68’s statistical power for survival endpoints is diminished by its early termination and small sample size. Limiting causal inference, OSERO is observational but prospective. SEQRT2 and the Ahn et al. study, on the other hand, offer real-world generalisability; yet, they are both retrospective and more likely to have confounding and selection bias. Despite being retrospective, the ReTrITA study provides the strongest evidence for R/T because of its sizable sample size and steady benefit across a variety of outcomes and subgroups. The lack of bevacizumab combinations in ReTrITA, in contrast to the other three studies, may have limited its applicability to the current standard-of-care regimens, which increasingly use anti-VEGF drugs. The evidence as a whole is still complex. Three sizable real-world cohorts—SEQRT2, the Ahn et al. study, and, particularly, ReTrITA—indicate a trend or definite advantage in overall survival and safety with the R/T sequence, despite the PRODIGE 68 trial’s support for T/R for improved treatment practicality. These results highlight how crucial it is to tailor treatment choices to each patient’s unique circumstances, including performance status, past toxicities, and treatment objectives [33]. To conclusively determine the ideal sequence of R and T in refractory mCRC, future randomised trials with adequate power and the inclusion of contemporary combos (such as T + bevacizumab) are necessary.

ReTrITA’s subgroup analysis showed that R/T was more beneficial for a variety of patient categories, which the other studies could not state as clearly. Although OSERO differentiated outcomes by tumour sidedness and molecular status, neither the Ahn et al. study nor SEQRT2 was able to identify subgroup-based differences with enough granularity. The ReTrITA results are stronger and more applicable in practice since they are consistent across subgroups. Given that they address a number of issues that come up in routine clinical practice and offer recommendations for what to do or not do following challenge therapy, our findings may be highly intriguing with regard to the continuum of care for patients with mCRC. This is especially true for patients with an ECOG PS = 2, patients with distinct metastatic patterns, male patients versus female patients, and other variables. Confirming our results with prospective trials with randomised controls would be both desired and justified. The fact that our study includes a population of patients that is representative of those encountered in routine clinical settings makes our analysis more therapeutically useful, which is one of its primary benefits.

However, the ReTrITA study has intrinsic drawbacks, the main one being its non-randomised and retrospective design, which raises the possibility of unmeasured confounding and selection bias. Furthermore, the comparison with recent trials like SUNLIGHT, which showed better results for T in combination with bevacizumab, may be limited by the lack of a central radiologic evaluation, inconsistent dose-escalation approaches (such as the ReDOS strategy), and exclusion of bevacizumab. Given the growing importance of antiangiogenic combinations in late-line conditions, the results of ReTrITA underscore the need for pragmatic randomised trials or prospective real-world investigations comparing sequential R/T vs. T + bevacizumab followed by R and vice versa. Furthermore, head-to-head comparisons between R, T, and fruquintinib—either as monotherapy or within rational sequences—will be crucial to defining optimal treatment algorithms for refractory mCRC as the drug becomes more widely available in Western nations after being approved in Asia and its recent authorisation by the FDA.

To sum up, ReTrITA not only validates the effectiveness of trifluridine/tipiracil and regorafenib as salvage treatments but also offers strong support for the R/T sequence in terms of feasibility and survival. These findings highlight the value of real-world data in guiding treatment decisions in the changing environment of mCRC and support a regorafenib-first approach for suitable patients.

## 6. Conclusions

The ReTrITA study provides one of the largest real-world evaluations of regorafenib and trifluridine/tipiracil, both as monotherapy and in combination, in heavily pretreated patients with metastatic colorectal cancer. Both overall and progression-free survival were statistically significantly improved by sequentially administering regorafenib and trifluridine/tipiracil (R/T) as opposed to the reverse sequence (T/R), even though no significant differences were seen between the two agents when used separately. These results were consistent with preceding real-world data and across important clinical subgroups.

Unlike experimental treatments, T and R are already authorised and reimbursed in numerous countries. Because of this, the R/T approach is scalable in a variety of healthcare settings, instantly implementable, and cost-neutral when compared to more recent biologics. The significance of therapy tolerance and sequencing feasibility in standard practice was further emphasised by this study. Despite being retroactive in nature, ReTrITA offers compelling evidence supporting the use of a salvage therapy strategy that prioritises regorafenib. The prospective validation of these results, investigation of optimised sequences, including combination regimens like trifluridine/tipiracil with bevacizumab, and comparative trials that include fruquintinib in the evolving treatment landscape are all necessary for future research.

## Figures and Tables

**Figure 1 cancers-17-02037-f001:**
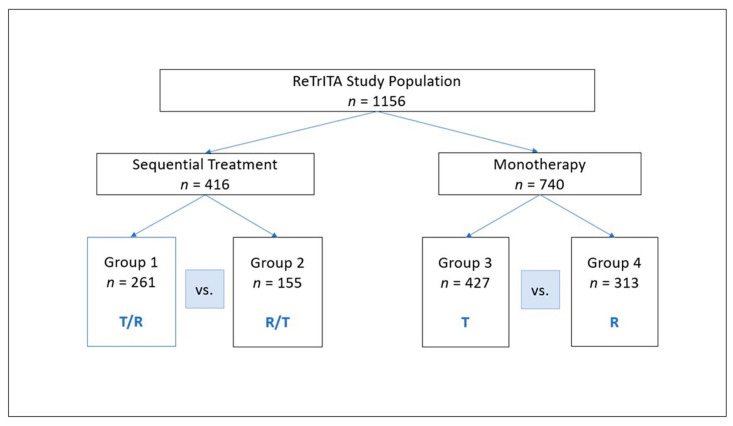
ReTrITA study design. Abbreviations: T, trifluridine/tipiracil; R, regorafenib.

**Figure 2 cancers-17-02037-f002:**
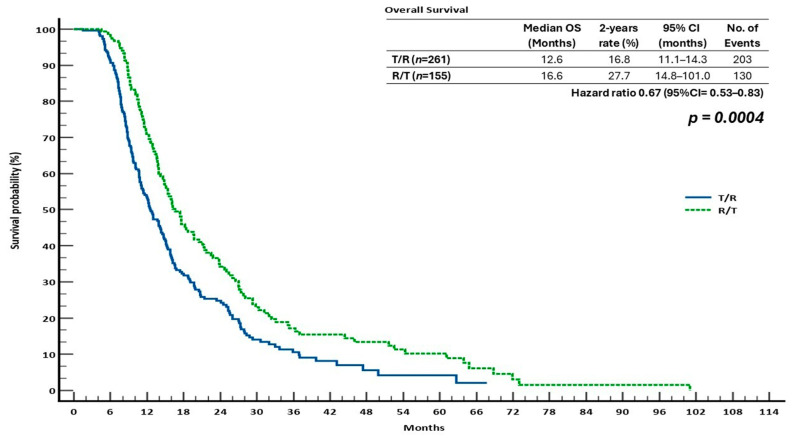
Kaplan–Meier overall survival (OS) graphs showing the differences between treating patients with metastatic colorectal cancer (mCRC) in the ReTrITA study sequentially with trifluridine/tipiracil followed by regorafenib (T/R) versus regorafenib followed by trifluridine/tipiracil (R/T).

**Figure 3 cancers-17-02037-f003:**
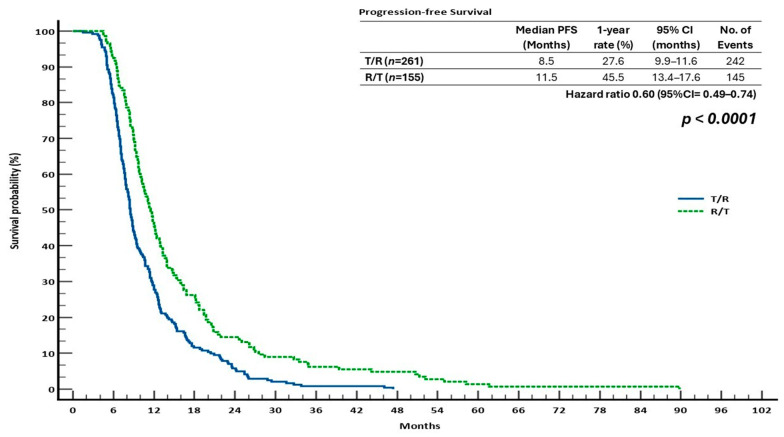
Progression-free survival (PFS) curves comparing sequential T/R and R/T treatments using the Kaplan–Meier method.

**Figure 4 cancers-17-02037-f004:**
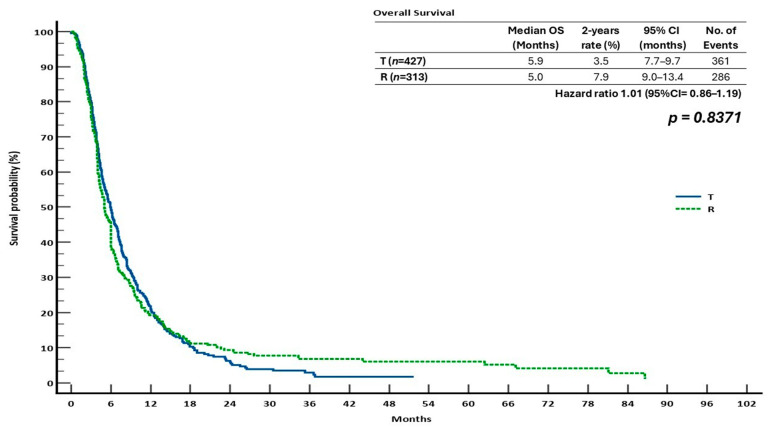
Kaplan–Meier curves for overall survival (OS), comparing the cohorts receiving monotherapy.

**Figure 5 cancers-17-02037-f005:**
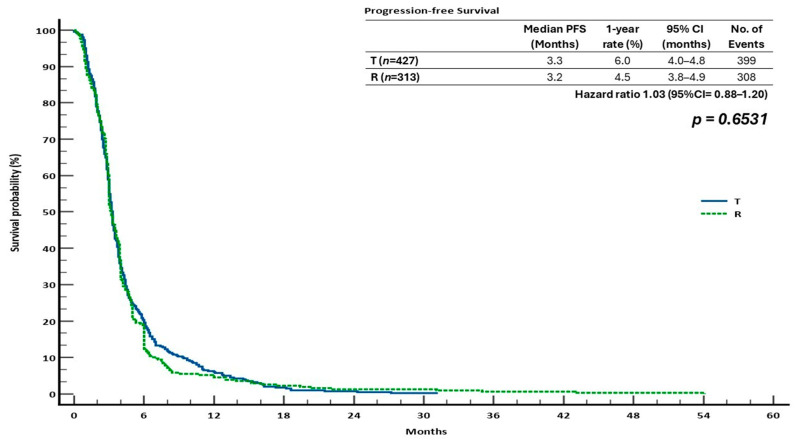
Kaplan–Meier analysis of the monotherapy cohorts’ progression-free survival (PFS).

**Figure 6 cancers-17-02037-f006:**
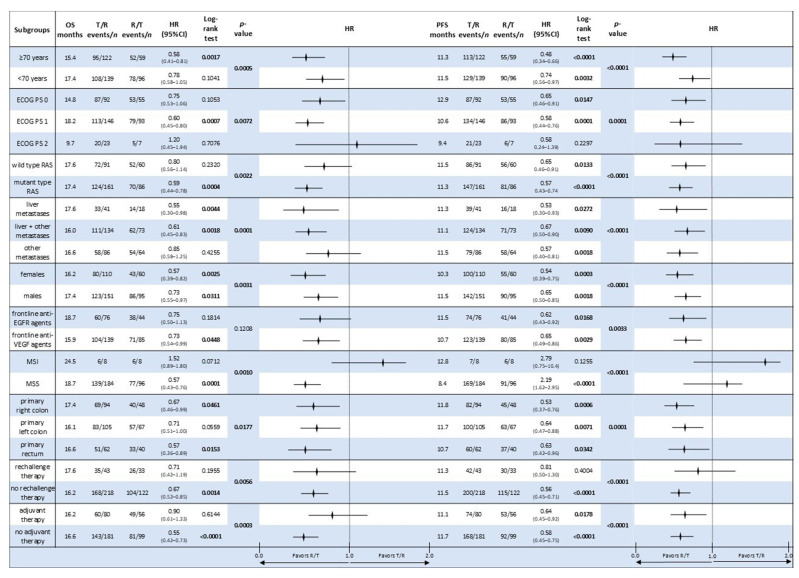
A forest plot showing subgroup analyses for overall survival (OS) and progression-free survival (PFS) in metastatic colorectal cancer patients treated with sequential T/R versus R/T. Statistically significant *p*-values are reported in bold. Abbreviations: OS, overall survival; PFS, progression-free survival; HR, hazard ratio; PS, performance status; T, trifluridine/tipiracil; R, regorafenib; CI, confidence interval; and n, number. The bold numbers in the table indicate statistically significant *p*-values.

**Figure 7 cancers-17-02037-f007:**
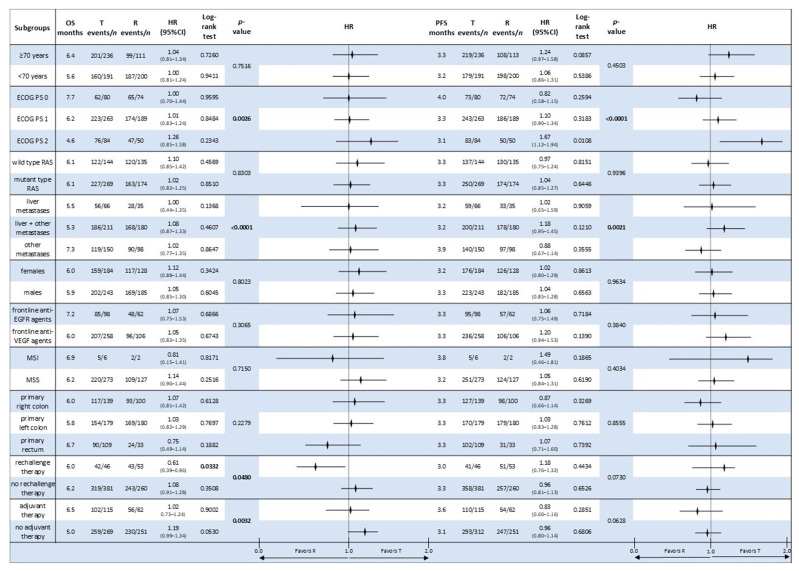
A forest plot showing subgroup analyses for overall survival (OS) and progression-free survival (PFS) for patients receiving non-sequential trifluridine/tipiracil (T) or regorafenib (R) in the ReTrITA study. Statistically significant *p*-values are reported in bold. Abbreviations: OS, overall survival; PFS, progression-free survival; HR, hazard ratio; PS, performance status; T, trifluridine/tipiracil; R, regorafenib; CI, confidence interval; and n, number. The bold numbers in the table indicate statistically significant *p*-values.

**Table 1 cancers-17-02037-t001:** Baseline demographic and clinical characteristics of patients included in the ReTrITA study.

	T/R	R/T	*p*-Value	T	R	*p*-Value
	N (%)	N (%)		N (%)	N (%)	
Total	261 (100)	155 (100)	**<0.0001**	427 (100)	313 (100)	**<0.0001**
Age Median (min–max)	69 (86–34)	67 (85–44)	0.8638	71 (89–40)	65 (33–86)	0.6069
Age			0.0847			**<0.0001**
≥70 yrs	122 (46.7)	59 (38.1)		236 (55.3)	113 (36.1)	
<70 yrs	139 (53.3)	96 (61.9)		191 (44.7)	200 (63.9)	
Sex			0.4912			0.5502
Female	110 (42.1)	60 (38.7)		184 (43.1)	128 (40.9)	
Male	151 (57.9)	95 (61.3)		243 (56.9)	185 (59.1)	
RAS status			0.3205			**0.0119**
Wild type	91 (34.9)	60 (38.7)		144 (33.7)	135 (43.1)	
Mutant type	161 (61.7)	86 (55.5)		269 (63.1)	174 (55.6)	
Unknown	9 (3.4)	9 (5.8)		14 (3.2)	4 (1.3)	
Primary tumour location			0.5758			**<0.0001**
Right side	94 (36.0)	48 (31.0)		139 (32.5)	100 (32.0)	
Left side	105 (40.2)	67 (43.2)		179 (42.0)	180 (57.5)	
Rectum	62 (23.8)	40 (25.8)		109 (25.5)	33 (10.5)	
MMR			0.1670			**<0.0001**
MSI	8 (3.1)	8 (5.2)		6 (1.4)	2 (0.6)	
MMR	184 (70.5)	96 (61.9)		273 (63.9)	127 (40.6)	
Unknown	69 (26.4)	51 (32.9)		148 (34.7)	184 (58.8)	
PS ECOG			0.2504			0.1741
0	92 (35.2)	55 (35.5)		80 (18.7)	74 (23.6)	
1	146 (55.9)	93 (60.0)		263 (61.6)	189 60.4)	
2	23 (8.9)	7 (4.5)		84 (19.7)	50 (16.0)	
Prior adjuvant therapy			0.2501			**0.0022**
Yes	80 (30.7)	56 (36.1)		115 (29.9)	62 (19.8%)	
No	181 (69.3)	99 (63.9)		269 (70.1)	251 (80.2)	
Metastatic disease sites			0.1847			0.0658
Liver only	41 (15.7)	18 (11.6)		66 (15.5)	35 (11.2)	
Liver + other	134 (51.3)	73 (47.1)		211 (49.4)	180 (57.5)	
Others	86 (33.0)	64 (41.3)		150 (35.1)	98 (31.3)	
CT: 1° line regimen			0.4942			**<0.0001**
Monochemotherapy	18 (6.9)	6 (3.9)		9 (2.1)	1 (0.3)	
Doublet chemotherapy	215 (82.4)	128 (82.6)		344 (80.6)	168 (53.7)	
Triplet chemotherapy	19 7.3)	13 (8.4)		37 (8.7)	16 (5.1)	
Unknown	9 (3.4)	8 (5.2)		37 8.7)	128 (40.9)	
CT: 2° line regimen			0.9257			**<0.0001**
Monochemotherapy	24 (9.2)	13 (8.4)		48 (11.2)	17 (5.4)	
Doublet chemotherapy	201 77.0)	123 (79.4)		323 (75.6)	151 (48.2)	
Triplet chemotherapy	6 2.3)	4 (2.6)		8 (1.9)	7 (2.2)	
Unknown	30 (11.5)	15 (9.7)		48 (11.2)	138 (44.1)	
Rechallenge therapy			0.2197			**0.0151**
yes	43 (16.5)	33 (21.3)		46 (10.8)	53 (16.9)	
no	218 (83.5)	122 (78.7)		381 (89.2)	260 (83.1)	
Biological agents: 1°line			0.9494			**<0.0001**
Anti-EGFR use	76 (29.1)	44 (28.4)		98 (23.0)	62 (19.8)	
Anti-VEGF use	139 (53.3)	85 (54.8)		258 (60.4)	106 (33.9)	
None	46 (17.6)	26 (16.8)		71 (16.6)	145 (46.3)	
Biological agents: 2°line			0.1836			**<0.0001**
Anti-EGFR use	16 (6.1)	10 (6.5)		27 (6.3)	14 (4.5)	
Anti-VEGF use	157 (60.2)	106 (68.4)		249 (58.3)	120 (38.3)	
None	88 (33.7)	39 (25.2)		151 (35.4)	179 (57.2)	

Abbreviations: T, trifluridine/tipiracil; R, regorafenib; MSI, microsatellite instability; MMR, mismatch repair; PS, performance status; and CT, chemotherapy. The bold numbers in the table indicate statistically significant *p*-values.

**Table 2 cancers-17-02037-t002:** Efficacy outcomes according to the treatment group. The bold numbers in the table indicate statistically significant *p*-values.

	OS					PFS					ORR		DCR	
	mOS (mos)	2y-OS (%)	3y-OS (%)	HR (95% CI)	*p*-Value	mPFS (mos)	1y-PFS	2y-PFS	HR (95% CI)	*p*-Value	PR + CR (%)	*p*-Value	PR + CR +SD (%)	*p*-Value
**T/R**	12.6	16.8	5.3	0.67 (0.53–0.83)	**0.0004**	8.5	27.6	5.7	0.60 (0.49–0.74)	**<0.0001**	3.4	1.0	32.9	0.9564
**R/T**	16.6	27.7	12.9			11.5	45.5	14.4			5.5		52.1	
**T**	5.9	3.5	1.1	1.01 (0.86–1.19)	0.8371	3.3	6.0	0.7	1.03 (0.88–1.20)	0.6531	2.8	0.2100	23.5	**<0.0001**
**R**	5.0	7.9	4.4			3.2	4.5	1.2			2.1		22.9	

**Table 3 cancers-17-02037-t003:** The incidence of grade 3–4 adverse events (AEs), stratified by treatment cohort.

	T/R		R/T			T		R		
	*n*	%	*n*	%	*p*-Value	*n*	%	*n*	%	*p*-Value
All G3/G4 events	242	100	184	100	**0.0050**	234	100	155	100	**0.0001**
All patients who experienced G3/G4 toxicities	145	55.5	107	69.0	**0.0167**	177	41.4	117	37.3	**0.0005**
All haematologic G3/G4 events	124	51.2	85	46.2	**0.0070**	182	77.8	13	8.4	**<0.0001**
All non-haematologic G3/G4 events	118	48.8	99	53.8	0.1971	52	22.2	142	91.6	**<0.0001**
Most common haematologic G3/G4 toxicities					0.9385					**0.0336**
Neutropenia	90	72.6	59	69.4		127	69.8	4	30.8	
Febrile neutropenia	2	1.6	1	1.2		1	0.5	0	0.0	
Thrombocytopenia	7	5.6	5	5.9		6	3.3	1	7.7	
Anaemia	25	20.2	20	23.5		48	26.4	8	61.5	
Most common non-haematologic G3/G4 toxicities					**0.0057**					**<0.0001**
Fatigue	47	39.8	31	31.3		28	53.8	33	23.2	
Hand–foot skin reaction	11	9.3	26	26.3		0	0.0	27	19.0	
Hypertension	6	5.1	11	11.1		1	1.9	14	9.9	
Liver dysfunctions	4	3.4	4	4.0		1	1.9	5	3.5	
Diarrhea	17	14.4	6	6.1		10	19.2	13	9.2	
Skin disorders	2	1.7	3	3.0		0	0.0	11	7.7	
Others	31	26.3	18	18.2		12	23.1	39	27.5	

Statistically significant *p*-values are reported in bold. Statistical comparisons (*p*-values) are presented for all event rates, highlighting the different safety profiles of T and R as stand-alone treatments and when used sequentially.

## Data Availability

The data to support the results reported in this study are available from the corresponding author upon reasonable request.

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
