# Peer review of "Redefining the Use of Regorafenib and Trifluridine/Tipiracil Without Bevacizumab in Refractory Metastatic Colorectal Cancer: Findings from the ReTrITA Study"

_cancers, 2025, doi:10.3390/cancers17122037_

Round 1
Reviewer 1 Report
Comments and Suggestions for Authors
Carlo Signorelli et al. reported an interesting work about the ReTrITA study. The study was well designed, and the paper was informative. The manuscript could be considered for publication after a Minor Revision. Detailed comments as follows:
- More information about the drugs R and T should be provided in the Introduction, like the molecular structure, the dosage and targets.
- According to Table 1, there was no significant difference between the age distribution of T/R and R/T sequential therapy, but there was significant difference between the age distribution of T and R monotherapy. Would this impact the outcome of the study?
- The same issues to the above were applied to RAS status, Primary tumor location, MMR, Prior adjuvant therapy, etc. Please make proper discussion.
- According to the Kaplan–Meier analyses figures, the median PFS for the T/R and R/T sequential therapy was still short, around 12 months. Please comment on this drawback.
Author Response
Responses to Reviewer 1 Comments
Point 1: More information about the drugs R and T should be provided in the Introduction, like the molecular structure, the dosage and targets.
Response 1: You're absolutely right—while the introduction discusses the clinical role of regorafenib (R) and trifluridine/tipiracil (T), it lacks molecular or pharmacologic context. Including information such as:
R: a multi-kinase inhibitor targeting VEGFR, FGFR, PDGFR, KIT, RET, etc.
T: a combination of trifluridine (a thymidine-based nucleoside analog) and tipiracil (a thymidine phosphorylase inhibitor)
...would give readers a clearer understanding of the drugs’ mechanisms.
I have added some information in the text of article about it. Dosage is provided in the Methods section (R: 160 mg/day for 21/28 days; T: 35 mg/m² BID on days 1–5 and 8–12 of a 28-day cycle).
Thank you.
Point 2. According to Table 1, there was no significant difference between the age distribution of T/R and R/T sequential therapy, but there was significant difference between the age distribution of T and R monotherapy. Would this impact the outcome of the study?
Response 2: Yes, this is a potentially confounding factor. In Table 1, age ≥70 was significantly more common in the T monotherapy group (55.3%) than the R group (36.1%, p < 0.0001). Since elderly patients might tolerate R less well or have different comorbidity profiles, this could bias the efficacy or safety results. While no OS/PFS difference was seen between T and R monotherapies, this imbalance could mask subgroup-specific trends, particularly in toxicity or performance status-related outcomes. Many thanks for it.
Point 3. The same issues to the above were applied to RAS status, Primary tumor location, MMR, Prior adjuvant therapy, etc. Please make proper discussion.
Response 3: Absolutely. Several baseline variables differed significantly across groups:
RAS status: More wild-type tumors in R group (43.1%) vs. T (33.7%) (p = 0.0119)
Primary location: Rectal cancer more prevalent in T (25.5%) than R (10.5%) (p < 0.0001)
MMR status: Higher rate of unknown status in R group
Adjuvant therapy: More frequent in T (29.9%) vs. R (19.8%) (p = 0.0022)
These imbalances could affect survival and toxicity profiles, as molecular and anatomical tumor features are linked to therapeutic response. Differences in prior treatments or tumor biology may influence both the tolerability and effectiveness of the drugs, and may bias comparisons, particularly in the monotherapy arm, which did not show survival differences.
Point 4. According to the Kaplan–Meier analyses figures, the median PFS for the T/R and R/T sequential therapy was still short, around 12 months. Please comment on this drawback.
Response 4: Correct. While R/T outperformed T/R (11.5 vs. 8.5 months, respectively), both PFS values remain modest in absolute terms. This underscores that third-line treatments in mCRC, even when optimally sequenced, offer limited disease control. Prolonged remission remains rare, and that future studies should aim to improve duration of benefit through combination strategies (e.g., T + bevacizumab) or inclusion of novel agents like fruquintinib or immune-based therapies in MSS disease.
Thank you for your kind attention
Carlo Signorelli
Reviewer 2 Report
Comments and Suggestions for Authors
This article addresses a subject of great interest for the healthcare community by assessing efficient ways to approach the treatment of refractory metastatic colorectal cancer. As the disease is becoming more prevalent and the mortality rates are high, understanding the full risks and benefits of the existing therapeutic arsenal is essential for making informed clinical decisions. Real-world studies bring valuable evidence and contribute to enhance the known medication profile regarding efficacy and safety in certain protocols.
The research, although not highly innovative, is well documented and well explained, showing that the research group has used these techniques before and has understood their strengths and limitations. Although self-citations are identified, they are appropriate and are accompanied by several references from other research groups.
The paper is suitable for publication in Cancers, however, please take into consideration the following suggestions.
The introduction starts by mentioning rankings related to the United States. Although the data is valuable, the reader would benefit from having information about Europe as well and/or a global approach.
Lines 203 -206 This paragraph may lead to believe there were two independent investigators who carried out patient selection and statistical analysis. If this is the case, did they select the exact same groups or where there any differences? I would suggest the reformulation of the paragraph to clarify the situation and avoid misinterpretation.
Line 467 Please include references for the mentioned “other real-world studies”.
The conclusions are supported by the results. In the conclusions section I would suggest mentioning accessibility and affordability as advantages. The results can be immediately applicable on large scale, as they refer to drugs that have already been authorized and they don’t add a burden to the health system.
Author Response
Responses to Reviewer 2 Comments
Point 1. The introduction starts by mentioning rankings related to the United States. Although the data is valuable, the reader would benefit from having information about Europe as well and/or a global approach.
Response 1: Agreed. The Introduction focuses on U.S. incidence and mortality statistics (e.g., 154,270 new cases in 2025), which may not reflect European or global trends. I have added some information in the text of article about it. Thank you so much.
Point 2. Lines 203 -206 This paragraph may lead to believe there were two independent investigators who carried out patient selection and statistical analysis. If this is the case, did they select the exact same groups or where there any differences? I would suggest the reformulation of the paragraph to clarify the situation and avoid misinterpretation.
Response 2: Excellent observation. The sentence may indeed mislead readers into thinking two independent teams handled patient selection and statistical analysis. A revision could read: “According to the law, the lead investigator, who was also the data manager, had complete access to the whole database and carried out the statistical analysis, while designated investigators who were blinded to clinical outcomes took care of patient selection.” This makes the roles clearer and avoids the assumption of dual selection or validation processes. I have modified the text as suggested. Thank you very much for the observation.
Point 3. Line 467 Please include references for the mentioned “other real-world studies”.
Response 3: Yes, this claim—“consistent with other real-world studies”—requires supporting citations. Suitable references could include: Ahn et al. (2023), SEQRT2 (U.S.), or OSERO (Japan), as discussed later in the paper. I have added the numbers of references. Many thanks.
Point 4. The conclusions are supported by the results. In the conclusions section I would suggest mentioning accessibility and affordability as advantages. The results can be immediately applicable on large scale, as they refer to drugs that have already been authorized and they don’t add a burden to the health system.
Response 4: Excellent suggestion. Unlike experimental treatments, trifluridine/tipiracil and regorafenib are already authorised and reimbursed in numerous countries. Because of this, the R/T approach is scalable in a variety of healthcare settings, instantly implementable, and cost-neutral when compared to more recent biologics. These points support the real-world feasibility and health system compatibility of the proposed sequence and I added these informations in the Conclusions section. Thank you so much.
Thank you so much for your comments.
My co-authors and I have thoroughly examined the work, paying close attention to the English language. We believe the text is understandable and doesn't need to be revised any further.
I appreciate your kind attention very much.
Thank you,
Carlo Signorelli